# Governmental Response to 'COVID-19' and Religious Freedom in Korea as Compared to the United States

Daeho Choi [1] and Taesoo Kim [2,*]

1   Department of Law and Public Service, Daejin University, Pocheon 11159, Republic of Korea
2   Daesoon Academy of Sciences, Daejin University, Pocheon 11159, Republic of Korea
*   Correspondence: tskim1003@daejin.ac.kr

**Abstract:** During the COVID-19 crisis, the Korean government's restrictions on religious freedom have caused several reactions. These can be divided into two categories: first, as several Christian groups have become the center of controversy over exacerbating the spread of COVID-19, the public sentiment towards religious gatherings has lost favorability. Second, there had been an increase in resistance to Christian groups and their right to worship. Regarding these issues, although the resistance from religious groups has not been as prominent as in the West, it is believed by both Korea and the West that the conflict between state power and religious right is at the root of this problem. This study reviews the Korean government's restrictions on religious activities and the consequent resistance of churches from a legal and institutional angle. In addition, this study compares the Korean approach to that of the United States in terms of governmental measures, the resistance of churches, and judgments issued by the judiciary. After demonstrating stronger restrictions in the case of Korea, this study evaluates this response in terms of legal and religious pluralism, and suggests that more sophisticated legal and institutional supplementation is needed to establish a robust and viable religious governance based on diversity and mutual respect.

**Keywords:** COVID-19; religious liberty; freedom of religion; governmental measures; legal restrictions; worship; epidemic prevention; religious organizations; prohibition of gatherings





## 1. Introduction

The ongoing response to COVID-19 has left everyone with little recourse aside from living lives of great uncertainty, endangered civic freedom, reduced autonomy, and indefinite future prospects. Although almost three years have passed since the outbreak of COVID-19, the crisis does not show any signs of termination, and the possibility that a future pandemic could lead to forced compliance with renewed or even heightened levels of regulation lingers uncomfortably in the background.

In South Korea (hereinafter, Korea), from the beginning of the pandemic, there was a tacit recognition that the state could implement strong measures when justifiable, and people would be expected to abide by the regulations in order to save people's lives and protect their health from that hitherto unknown plague. The restrictions on daily living due to the crisis occur in many ways, but they also have a significant impact on religious life, especially Christian life. Church services were replaced by online worship services, and group Bible studies and worship activities were rendered nearly impossible. These restrictions on religious activities were accelerated by alleged refusal to quarantine, false reports, political demonstrations by specific religious groups, such as the *Shincheonji* Church of Jesus, which was suspected of triggering the spread of COVID-19. The public antipathy towards these Christian and new religious groups provided the Korean government with the perfect justification to implement strong regulations based on the "Infectious Disease Control and Prevention Act" (hereinafter, IDCP Act).

In light of the scientific fact that infectious diseases are transmitted by person-to-person contact or droplets, there has been a tacit recognition that social gatherings, whether

religious or not, may need to be temporarily suspended. However, the following question was raised as to whether it would be really desirable to suspend religious activities uniformly without sufficient micro-adjustments according to the nature, contents, and methods of the gathering—to what extent can a state impose restrictions on religious freedom by executive order to prevent the massive spread of an epidemic? The troublesome problem is that, while religious freedom is guaranteed under Article 20 of the Constitution of Korea, restrictions are also possible under Article 37 (2) of that same Constitution. Given that reality, how can religious freedom be guaranteed while setting limits?

Before a full-fledged discussion, first the concepts of "religious freedom" and "religious liberty", as used in this study, should be outlined. In most cases, "religious freedom" and "religious liberty" are interchangeable. Yet, utilizing the social contract theory and legal perspectives in line with the natural law tradition[1], this study tends to use 'freedom' in a more limited sense to imply the inalienable, ideal state of freedom that humans innately enjoy from a state of nature prior to the existence of the state or of laws'.[2] Freedom of conscience, will or action fall under this category. On the other hand, 'liberty' in the natural law tradition, is used to refer to legal rights based on autonomy formed after the introduction of a social contract or laws to which individuals cede power to secure their protection.[3] In the modern context, these concepts are mainly used to judge whether a specific act of an individual or the state violates basic human rights. Additionally, human autonomy refers to reason that can control decisions and actions by oneself; not by external coercion. From this perspective, this study tends to use the term 'religious liberty' in the sense that it presupposes the citizen's potential for autonomous religious actions within a legal national community.

Based on this conceptual definition, in order to reconcile the problems raised above, this study reviews the Korean government's restrictions on church activities and the consequent resistance of Christian groups, mainly from a legal and institutional point of view. Yet, when necessary, legal and religious hermeneutical perspectives are also used. To this end, after a brief overview of the relationship between religion and the state in Korea, this study elaborates upon the government's countermeasures to prevent the spread of the pandemic and the resulting restrictions on religious organizations. The main focus will be on several Christian and new religious groups, which were regarded as main contributors to the spread of COVID-19. In addition, this study also examines various factors that affect public resistance and acceptance in response to the governmental restrictions on these groups.

Further, for comparison, this study reviews the United States' (hereinafter, U.S.) state actions on the same issue, as well as the consequent resistance of churches and representative judicial judgments. Finally, it concludes by suggesting the need to reconcile the exercise of state power and religious freedom in Korea by improving legal-institutional norms and practices, and by promoting politico-religious cooperation based on a pluralistic model of religious governance.

Lastly, one limitation of this study that should be stated is that the sociological discussion of religion has not been sufficiently reviewed, and therefore, this thesis focuses on legal reflections on governmental restrictions on religious activities via the comparison of South Korea and the U.S.

## 2. A Point of Departure: The Relationship between the State and Religion

Fitzgerald explains how "Locke distinguishes religion from politics and government as an essential move toward liberalism (and later neoliberalism). A key aspect of this reconfiguration of religion was its privatization as faith in the inner conscience of the individual, thus radically distinguishing it from the public domain of rational governance and from scientific knowledge. Likewise, Locke authorized a new discourse on religion, as well as a new understanding of governance and the political state as non-religious" (Stack et al. 2015, p. 14).

Most countries have struggled with the question of how to build the relationship between the state and religion on the basis of constitutional law. The system of separation of church and state, which was created in the course of both the American Revolution of 1797 and the French Revolution of 1789, has historically manifested in various forms, from the peaceful coexistence of various religions to outright oppression and hostility against all religions. Still, in most countries, including the U.S., issues surrounding the 'separation of religion (from the state)' and 'religious freedom' occur frequently, even though ① there is no official state religion and ② religious freedom is guaranteed by constitutional law (Landsberg and Jacobs 2007, p. 179).

With regard to this issue, there are several models for the relationship between the state and religion within civil societies: (1) A confrontational structure between the two (2) A symbiotic cooperative structure consisting of the state's protection of religion within civil society, and the voluntary cooperation of religion with the state; (3) Various models, including religious governance, whereby the voices of civil society are reflected in the state through consultative bodies made up of government agencies, local governments, civic groups, interest groups, as well as religious groups and lay service organizations, such that the state can converge the opinions of religious organizations in connection with civil society. Under this model, religious groups can effectively influence state actions or policy decisions by utilizing these consultative bodies made up of various forms and channels. In general, mature democratic societies aim for Models 2 and 3.

Meanwhile, when talking about 'religious liberty' within these models, in most cases, liberty and freedom can be used interchangeably. Yet, etymologically speaking, there are many controversies about the concepts of liberty and freedom, the former being of Latin origin and the latter being of Germanic and Anglo-Saxon origins. It is generally argued that the two words have developed in parallel and then merged after the Norman invasion of England. Despite these differences, there are also many claims that these two terms have been developed in close association with the ancient and medieval master–slave relationship in mind. In particular, liberty was initially regarded as the least liberated state in which there is no owner, and there are the free movement of the body, procreation, labor, and leisure activities (Pitkin 1988, pp. 529–31, 541). Then, with the emergence of the modern civil society and state, it began to be regarded as free participation in civic activities of democratic communities under legal protection.

Noting the historical developments of the concepts of liberty and freedom, in this study, 'religious liberty' is mainly used in a legal context, e.g., as a legal right based on human autonomy; that is, a free condition enjoyed by citizens in everyday life, and also enjoyed within legal and normative scopes. In particular, this study deals with situations of national crises like the pandemic, where the conditions of Models (1) and (2) become stronger. In other words, religious liberty used here is a sort of negative concept, meaning 'a basic right with the possibility of being restricted by state power for the public good'.

Along this line, the 'liberty' in the American Declaration of Independence was used following the spirit of the French Revolution; heralding the modern human liberation departed from the structure of domination and subjugation of ancient times. On the other hand, 'freedom' can be used in a positive sense, implying an ideal state, when it is said that 'human beings in the state of nature are free', and this assumes that the state of nature is an ideal state without oppression. Therefore, religious freedom means the inherent natural freedom that has not been transferred to the state—such as freedom of religious conscience and freedom of belief. Accordingly, the state cannot intervene in religious freedom as described in this positivistic sense. On the other hand, religious liberty is understood to have the dual aspect of escape from infringement by state power while being guaranteed by state power. Overall, the 'concept of religious liberty' can be understood as ultimately seeking the state in which institutional guarantees are relatively well secured, while aiming for the stage of Model 3. At this stage, political participation is actively sought, such as religious governance or the exercise of civil rights.

Meanwhile, in relation to this point, the act of resistance by religious groups for the protection of their right to live and for their own interests can also be seen as an effort to move from the state described by Model 1 to that of Model 2 or 3 by opposing the limited liberty of autonomous agency bound by governmental regulations. This, of course, includes the objection to the case that the state abandons the obligation to affirmatively protect religious liberty and human rights. With these efforts reflected, Model 3 is considered close to the ideal state in which political and civil action is actively sought to secure an institutionalized relationship with the state. In the religious sphere within civil society, Model 3 also connotes an ideal state in which a religion meets the requirements of the state and civil society through mutual cooperation in accordance with the principle of reasonable reciprocity.

In accordance with this principle, religious organizations strive to more stably secure the religious liberty stipulated in the Korean Constitution through voluntary cooperation with national policies. However, this religious liberty was limited to the purpose of public benefit when confronting the COVID-19 pandemic. Even so, the restrictions of state power on religious liberty have certain limits, and accordingly, the legitimacy of religious liberty under state power is in question.

### 2.1. The Meaning of Religious Freedom and Its Implications in the Korean Constitution

Freedom of religion, along with freedom of conscience and freedom of body (bodily autonomy), is one of the oldest fundamental rights in the history of human rights. As Fitzgerald argues, historically, "Locke naturalized a very particular notion of 'freedom', articulated most clearly in liberalism and adopted as the raison d'etre of government in much of the world. Further, Locke's influence on Jefferson and much of the founding elite in drafting US Constitution ensured that Locke's division of religion and politics would become central not just to government but to the emergent capitalist system, whereby politics was to stand for the naturalized interests of the propertied class" (Stack et al. 2015, p. 14). This tradition of liberal democracy, based on a free market system, finds its ethical implications in the freedom of religion and conscience, guaranteed in the Virginia Bill of Rights (Article 16) and the French Declaration of Human Rights (Article 10). Since then, in most democracies today, it is a fundamental right that protects the dignity of the person and the free expression of the individual in the spiritual realm.[4]

In Article 20 (1), the Korean Constitution guarantees freedom of religion as follows: "All citizens shall have freedom of religion". As the Korean Constitution guarantees freedom of religion, the relationship between the state and religion is expected to be established in the following way: the constitutional guarantee of religious freedom and the resulting religious pluralism require neutrality from the state and tolerance from society. In due course, the state must maintain neutrality with respect to various religions, society is required to tolerate various religious truths, and religious organizations and their members must follow national law and ordinances while cooperating with the other sectors of civil society and the state.

### 2.2. The Guarantee of Religious Freedom
2.2.1. Protective Values for Religious Freedom

As briefly outlined, the meaning of constitutionally guaranteed religious freedom above shows that in the liberal democratic tradition, it is necessary for the state to value and protect religious freedom (Schutzgut). Yet, this question also relates to the meaning and content of the concepts of 'faith', 'conscience' and 'worldview'. Although it is difficult to define each concept distinctively, we can refer to their common usage in religious hermeneutics and the sociology of religion. In the European sociology of religion and religious hermeneutics, for instance, 'faith' and 'conscience' usually assume a religious tinge, and 'worldview' represents an 'Oberbegriff' that includes religion. Yet, in most cases, while religion acknowledges the existence of God, worldview does not acknowledge it, and the two are opposed in this respect (Mangolt/Klein/Starck 1985, S. 421). Religion

and worldviews in general are beliefs about the "human belief in the ultimate work (Überzeugungen des Menschen von den letzten Dingen)", and in particular, about the origin of the world and man's place therein, as well as the origin of human life and the meaning of death. (Ipsen 2014, p. 110).

What is decisive here is the substance of whatever can be considered 'truth (Wahrheit)' from the perspective of worldview. In this respect, freedom of religion (religion, worldview) guarantees the intrinsic ability of Man as a rational being, and the ability to recognize the world surrounding him in the name of his own conviction (Gewissheit). In other words, this points to 'truth (Wahrheit) in religion'. Here, the protective value of religious freedom is, after all, a conviction of the truth, different from other (scientific, for example) cognitions, and exempt from the search for evidence and 'objective fact (objektive Gegebenheiten)'. Therefore, the beliefs in question in religious freedom are objectively personal convictions based on mutually repulsive needs for truth (Wahrheitsanspruch), but because of the desire for truth arising from personal convictions, the protection of personal convictions necessarily entails the imperative of tolerance (Toleranzgebot) (Ipsen 2014, S. 110).

In this light, religious freedom is the right to defend against state sanctions (Sanktionen) and discrimination (Diskriminierungen) associated with a particular belief (Abwehrrechte) and the right to defend against state interference in matters of faith (von Münch and Mager 2014, S. 213). Likewise, since religious freedom belongs to the "private realm (Privatsphäre)", the state cannot intervene. However, if an action based on religious freedom transcends the limits of the private sphere and conflicts with the behavior of others in the "social sphere (Sozialsphäre)", an adjustment is necessary. Yet, in principle, this should only occur to that extent that obedience to the law is applicable to all (Ipsen 2014, S. 112).

### 2.2.2. Restrictions on Religious Freedom

With these issues in mind, the Korean case can now be examined. The provisions on freedom of religion in the Korean Constitution are composed as follows:

> The freedom of religion under Article 20 (1) of the Constitution generally consists of freedom of faith, freedom of confession of faith, and freedom of practice of faith (2000 Hun-M159)[5].

> Freedom of belief, the freedom to form and decide beliefs in the inner world, is absolutely protected because there is no possibility of conflict with other legal interests (2009 Hun-M529).

> On the other hand, freedom of confession of faith and freedom of realization of faith, which are the freedoms to realize one's faith in the outside world, may be restricted in accordance with Article 37 (2) of the Constitution for the protection of the public interest or the legal interests of a third party (2000 Hun-M159).

As quoted above, in the Korean Constitution, religious freedom can be subject to restrictions from the moment when inner faith is converted into externally expressed religious activity. In particular, freedom of religion may be restricted in terms of protecting the life and health of third parties. For example, certain churches may seek understanding from nearby residents to tolerate noise from worship within a certain range. However, it may be limited by frequency, time, and level of noise. Therefore, it is required for both parties to reach a reconciliation of their conflicting legal interests so that religious activities are possible while protecting the basic rights of others (Mangolt Klein·Starck 2014, S. 447).

### 2.3. Religious Neutrality of the State

#### 2.3.1. Denial of State Religions and the Principle of Separation of Politics and Religion

Article 20, Paragraph 2 of the Korean Constitution stipulates that "state religion is not recognized and religion and politics are separated", declaring the principle of refraining from establishing a state religion and maintaining the separation of church and state. Within this constitutional system that separates politics and religion, religious organizations

can enjoy basic rights and freedoms within the scope of the judicial system as a judicial organization, and they can function as large social organizations comparable to political parties and labor organizations. Accordingly, religious ideologies and forces existing in Korean society, like all other ideologies and forces within other pluralistic democratic societies, should have the right to participate in the open process of democratic decision-making and democratic interest management.

2.3.2. The Constitutional Meaning of Religious Neutrality

The state's duty to uphold religious neutrality is derived from a series of constitutional provisions, such as freedom of religion, the principle of refraining from establishing a state religion, the principle of the separation of church and state, freedom of conscience, and prohibition of discrimination on the basis of religion. A nation's worldview and religious neutrality are characteristic elements of a nation that seeks to support and guarantee a pluralistic and open society. A nation's worldview and religious neutrality are essential prerequisites for guaranteeing and realizing religious freedom, and this is essential for the nation's ability to function as a nation for all its citizens.

However, the request for religious neutrality does not presuppose that all matters related to religion be excluded from the national task area. When there is a clear concern about social harm from a particular faith community, it would not violate the request for neutrality for the state to take a critical attitude toward a particular faith community and warn the public of its dangers (2006 Da87903).

The state's duty of neutrality imposes obligations on the state to guarantee equal rights and opportunities in principle to all religions and worldviews. While it is prohibited for the state to give preference to a particular religion, the state may engage in justifiable discrimination under reasonable grounds in accordance with the principle of equality.

*2.4. The Spread of COVID-19 and the Characteristics of Some Chistian Groups*

In Korea, there have been a number of cases in which religious groups were recognized as having played a significant role in exacerbating the spread of COVID-19, and as a result, the public found it appropriate to question the degree of freedom of religious assembly that should be afforded to those groups. For example, the spread that occurred around a Liberation Day rally in 2020 centered on the Shincheonji Church in Daegu (MEDICAL Observer 2021) and Sarangjeil Church in Seoul. The members of both groups (Chosun Ilbo 2020) were considered to have played a decisive role in worsening the spread of COVID-19. Additionally, the high infection rate of religious groups such as the BTJ Sangju Center for the Nations (KBS NEWS 2021), which is another trend within the Christian lineage, as well as Intercop (The Hankook Ilbo 2021) and Yeongsaenggyo (The Hankyoreh 2021), also caused a ripple in society. In addition, there were large and small cases of infection, mainly from religious groups, such as a group infection that affected 72 people at Grace River Church in Seongnam.

Overall, these Christian groups were found to demonstrate many of the following characteristics:

- Religious viewpoints overwhelm reason and scientific viewpoints: Non-scientific cognition was a strong category in these Christian and new religious groups. Beliefs did not necessarily have to be scientific, but these groups promoted beliefs that faith could help overcome infectious diseases and spread medical misinformation. This likely led to high rates of infection.
- The practice of communal living: In some cases, congregates lived in groups that were blocked from the outside world, but there were also cases in which only members of the same church gathered without contact with outsiders. Their lives can be compared to secret societies or certain forms of esotericism to the extent that it was not easy to intervene or be involved from the outside.
- There was a strong tendency to understand the temple in the sense of physical 'place': In Christianity, there is a strong tendency to understand the temple as not being

confined to a specific place or location, but these groups believe in the contrary notion that their temple is a specific physical location. They attempted to prove their faith by gathering in those temples and worshiping within their holy spaces.

- There was a tendency to seek economic profits through religious gatherings: Some religious groups insisted on holding religious gatherings for donations. These religious groups are often criticized for having a weak sense of ethics, usually exhibited high levels of evangelism, and sometimes showed aggressive behavior in their promotional activities.

- Political positions also played a role: The weaker the recognition of the government's democratic legitimacy, the stronger that group's resistance to the government's quarantine measures would be.

As such, in the context of COVID-19, certain Christian and new religious groups resisted the government's quarantine measures and caused various social problems. As a result, social criticism against these groups grew stronger.

### 3. The Korean Government's Response to Religious Activities and Restrictions on Religious Freedom

From the early stages of COVID-19, the Korean government restricted or forbade performances, assemblies, ceremonies, and other gatherings to prevent the spread of the epidemic based on the IDCP Act. As a result, church services were replaced by online services and church facilities were closed. The following subsection details the main response of the Korean government to religious groups and this response especially affected the religious activities of Christian groups.

#### 3.1. The Progressive Spread of COVID-19 and the Government's Countermeasures

On 21 March 2020, the Korean government issued a statement saying, "It is urged that religious facilities, indoor sports facilities, and entertainment facilities with a high risk of group infection be suspended for a full week". On 8 July 2020, Prime Minister Jeong Sye-kyun, who attended a meeting at the Central Disaster and Safety Countermeasures Headquarters, announced that, unlike temples and cathedrals, core quarantine measures would be mandatory only for Christians. The government lifted the church quarantine measures on 22 July, two weeks after being accused of oppressing Christians due to this measure. The Korean government applied uniform standards to various facilities, such as restaurants, schools, and churches, despite their different characteristics. In cases wherein specific churches were found to have one or more confirmed cases of infection, the religious facilities in their entirety were called upon to collectively shutdown and this would also lead to a suspension of gatherings for all churches in the same area.

The Korean government raised the infectious disease crisis level to 'serious' on 23 February 2020, and the Central Disaster and Safety Countermeasures Headquarters was in charge of quarantine work [Korea Disease Control and Prevention Agency (hereinafter, KCDC 2021), 2 March 2021]. To prevent the spread of infectious diseases, testing, tracing, treatment (the 3T strategy) and social distancing adjustments were all implemented. Prevention of the spread of disease based on path movement checks and computerized management of quarantined persons, along with step-by-step quarantine measures was internally evaluated as significantly effective, and this approach was even branded as 'K-Quarantine' (Korean Policy Briefing 2020).

Unlike in some parts of the West, the Korean government's quarantine measures did not trigger a shutdown that blocked out the region and suspended a multitude of economic activities. Instead, in June 2020, a 'three-stage system' was prepared, and in November 2020, it was reorganized into a 'five-stage system'. In June 2021, a revised 'New 4th Stage of Distancing' was prepared, which simplified the existing measures and strengthened the autonomy of local governments. See Table 1 for details.

**Table 1.** Step-by-step quarantine rules for vulnerable facilities (Ministry of Health and Welfare 2021a).

| Division | Stage 1 | Stage 2 | Stage 3 | Stage 4 |
|---|---|---|---|---|
| Workplace | ▪ Staggered commuting, staggered lunch breaks, the urging of telecommunication | ▪ Businesses with 300 or more employees (excluding manufacturing): staggered commuting, staggered lunch breaks, proposal that 10% of meetings use telecommunication | ▪ Businesses with 50 or more employees (excluding manufacturing): staggered commuting, staggered lunch breaks, proposal that 20% of meetings use telecommunication | ▪ Staggered commuting for workplaces excluding manufacturing, staggered lunch breaks, proposal that 30% of meetings use telecommuting. |
| Religious facilities | ▪ 50% of total capacity | ▪ 30% of total capacity | ▪ 20% of total capacity | ▪ Only remote worship services, Masses, and legal meetings are allowed |
| | ▪ Refrain from gatherings, meals, and lodging | ▪ Ban on gatherings, meals, and lodging | | |

In addition, on 16 December 2021, after discussion with religious groups, such as Protestantism, Buddhism, and Catholicism, the government implemented the measures to strengthen the quarantine rules for religious facilities from 18 December 2021 to 2 January 2022. This was implemented for 16 days. See Table 2 for details.

**Table 2.** Limiting the number of participants and strengthening quarantine rules (Ministry of Health and Welfare 2021b).

| | |
|---|---|
| Regular religious activity | ○ All regular religious activities conducted under the supervision of religious facilities (religious people, religious groups, etc.) at a certain time and place.<br>○ When current vaccination status is unknown, participants should only reach 50% occupancy, and if all congregates are vaccinated, even 100% capacity is allowed. The number of participants is to be reduced to a maximum of 299, yet if it consists entirely of those who have completed vaccination, they can meet at 70% of the full capacity.<br>○ Basic quarantine rules, such as wearing a mask at all times, continue to apply. |
| Religious gathering | ○ Bible and scripture study, regional worship, preparation for missions or events, etc.<br>○ Currently, in the case of operating entirely with those who have completed vaccination, the range of private gatherings (6 people in the metropolitan area, 8 people in the non-metropolitan area) is possible, but in the future, it will be reduced to 4 people (nationwide), and only allowed in the case that all participants have been inoculated.<br>○ Small gatherings are limited to religious facilities, while eating and drinking and loud prayer are prohibited. |
| Religious event | ○ Prayer meetings, retreats, revival meetings, etc.<br>○ Currently, events with less than 100 people can be held regardless of vaccination status, and if there are more than 100 people, up to 499 people can be allowed, but only in the case that all participants are inoculated, but in the future, if there are less than 50 people, it will be possible with or without vaccination. In the case of more than 50, the maximum number is 299, and only permitted in the case of that all participants have been inoculated. |

**Table 2.** *Cont.*

| | |
|---|---|
| Etc. | ○　Choir and praise teams must be composed only of those who have completed vaccination to operate and they must wear masks at all times during activities.<br>○　Prohibition of acts that require people to take off their masks, such as eating and drinking in religious facilities, as well as acts that generate the oral ejection of spittle, such as group prayer and recitation at a high volume. |

*3.2. Court Attitudes Regarding Religious Freedom and Restrictions Thereof*

For the prevention of infectious diseases, the Korean government prohibited gatherings, suspended operations, and imposed restrictions on churches with confirmed cases of COVID-19 based on Article 49(3) of the IDCP Act by ordering them to close. In response, church groups filed a lawsuit asking the court to revoke the government's suspension on in-person worship. In this regard, the attitude of the Korean courts can be shown through representative examples.

3.2.1. Cases Upholding the Government's Suspension of Worship

①　Fact: On 18 July 2021, Sarang First Church held an in-person worship service with about 150 members, even though in-person worship was suspended due to the Seoul Metropolitan Government's order requiring 'social distancing in the metropolitan area'. In response to a request from the mayor of Seoul for administrative action, the Seongbuk-gu Office ordered Sarang First Church to "suspend its first operation" for 10 days. However, Sarang First Church continued in-person worship until 15 August. In response, the Seongbuk-gu Office ordered Sarang First Church to close the facility on the 19th for having proceeded with an in-person worship service during a period of suspension. On the 20th, the next day, a "secondary suspension order" was issued to have the church close their worship facilities until special measures were taken. Sarang First Church was dissatisfied with this and filed a lawsuit.

②　Contents of judgment: The applicant alleged that there was a risk of irreparable harm; that is, harm that would be impossible to recover from by means of financial compensation or that could not be tolerated or would be remarkably difficult to endure even with monetary compensation due to the closure of the facilities and the suspension on church activities such as worship. However, the church's application was dismissed in the following manner: "upholding the suspension that requires facility closure may have a significant impact on public welfare on the grounds that the need to defend public welfare as achieved through this is greater than the disadvantages that incurred by the applicant". (2021 Ah 12139)

3.2.2. Judgment against the Government's Suspension of Worship

①　Fact: In December 2020, the city of Seoul suspended in-person worship and allowed only remote (online teleconferencing, etc.) worship through the government's 2.5-step social distancing measures and special measures to strengthen quarantine during the year-end and New Year holidays, making it mandatory for religious facilities to follow quarantine rules. In response, church groups claimed that they did not pose a high risk of spreading COVID-19 if they held in-person worship services while complying with the government's quarantine rules. It was also argued that, in a situation where the Seoul Metropolitan Government has only taken measures to limit the number of people in other medium-risk facilities such as restaurants, it would be discriminatory to implement a de facto prohibition of gatherings for churches without providing a convincing rationale.

②　Court decision: The Seoul Metropolitan Government's actions were judged to be illegal (2021nu 76387) in that issuing a total suspension on in-person worship greatly infringes upon freedom of religion. Additionally, the disposition to suspend in-person worship in this case was found to have violated the principle of proportionality and

equality while also violating and abusing discretion. The specific reasons for this judgment were as follows:

- It is difficult to insist that the risk of spreading COVID-19 through in-person worship in churches is significantly higher than other facilities to such an extent that it warrants being blocked by prohibiting the gathering of church members. In addition, the Seoul Metropolitan Government's suspension of in-person worship did not meet the requirement for minimum infringement. This is because the risk of infection could be minimized within the range that society could tolerate by limiting the number of attendees and by prohibiting in-person gatherings and events within the churches other than worship.

- The church serves a positive function of supporting stable mental health by providing not only psychological comfort to the members but also a mental solution to overcome hatred for oneself and others. In the context of a gradual increase in the number of people complaining about a recession caused by long-term quarantine measures, there was no compelling reason to think that the functions these churches provide are inferior or less important than the essential production facilities. Therefore, no justifiable purpose could be found to discriminate against the churches but not restaurants.

- It could not be categorically stated that the risk of COVID-19 transmission would be significantly lowered through the simple suspension of in-person worship at churches, while allowing operations to continue in restaurants and other similar facilities by imposing quarantine rules such as obliging individuals to observe social distancing or certain specific business hours. In addition, it would not be possible to guarantee the ability to participate in worship services to members with limited internet access, or those who belonged to small churches, or faced difficulties in their livelihoods, or were of old age or lived with a disability, etc. Therefore, the discrimination being enacted was found to be unjustified and unnecessary.

- In turn, the proportionality requirement was not met due to the effect of inequality caused by discrimination. This is because there was too much restriction on the religious freedom of the plaintiffs, compared to the proportional degree of the public interest of protecting the health of the people, which was claimed to be achievable through a suspension of in-person worship.

- Additionally, remote worship could not be regarded as the same as in-person worship. That is, "Even if a church is fully capable of remote worship with enough human and physical facilities, there are procedures in Christian traditional worship that are not possible to practice through remote means. One example includes the sacrament, and according to Christian doctrine, participation in worship has an important religious significance. Church tradition also holds that worship is conducted according to a procedure consisting of sermons, praise, and prayers. Considering these aspects, it is difficult to claim that remote worship is the same as in-person worship".

*3.3. Review and Evaluation*

The spread of COVID-19 in Korea was sometimes linked to the religious activities of certain churches, and as a result, the government imposed very strong restrictions on religious activities. However, as the suspension of in-person worship continued for a long time, churches filed a lawsuit with the court, demanding that the government's suspension of in-person worship be lifted. It was decided that a total suspension of in-person worship infringed upon religious freedom for the following reasons: (1) According to the principle of excessive prohibition (proportionality), the government's quarantine measures against the churches should have been implemented in a way that would have minimized the violation of fundamental rights through other means of lowering the risk of transmission;

(2) It came short of sufficiently achieving the public interest of protecting the health of the people.

From the above precedent, we can evaluate that, unlike the case of Sarang First Church, the court sought a way to reconcile the conflict between the state and religion in society by minimizing the violation of fundamental rights, such as the risk of irreparable harm to the elderly or disabled. The court also declared the need for balanced quarantine measures according to the transmission rate while demonstrating respect for the weak and the values and roles of churches. However, Korea's case was more limited than that of the U.S. Supreme Court in that it did not apply to all believers but instead focused on the elderly and the disabled.

Moreover, in terms of legal and administrative measures for religious facilities and activities, considerable supplementary measures were required. For instance, the prohibition measures under Article 49, Paragraph 1 of the IDCP Act imposed too much administrative discretion, as they could impose a suspension rather than a restriction for the purpose of prevention. It ignored the step-by-step implementation of executive power in that the administrative power could choose the restrictions and prohibitions as desired without an objective criterion. These legal provisions violated the principle of proportionality, which states that limiting basic rights should be kept to a minimum. Due to the ambiguous character of these regulations, the head of one local government could only warn about the confirmed cases of COVID-19, while the heads of other local governments could limit gatherings, and the heads of some other local governments could issue suspensions on similar cases such that the administrative accountability would be too difficult to properly trace back to its sources. Likewise, when judging from the contents of these provisions, which were able to be arbitrarily interpreted for administrative convenience or interest in the legal context, the prohibition measures can be shown to be unconstitutional as they do not distinguish between restriction and prohibition. Therefore, it is desirable to make a clear distinction between these two measures and the following sanctions in detail. Of course, adding a warning step would have also been desirable.

For a better solution to complement the institutional deficiencies and governmental attitudes toward religions, in the next chapter, the case of U.S. federal and state level government will be examined with the response of the federal Supreme Court to the issue of religious freedom during COVID-19.

## 4. U.S. Response to COVID-19 and Religious Freedom

On 31 January 2020, the U.S Health and Human Services declared a public health emergency for the prevention of COVID-19. At that time, a decree was promulgated prohibiting religious activities and secular businesses.

Religious groups, arguing that the state's restrictions on religious activities violated the First Amendment[6] to freedom of religion, filed lawsuits with the courts demanding a suspension of the enforcement of related district ordinances and regulations. A review of the U.S. Supreme Court's precedent over the U.S. state government's response measures to religious activities will be provided below for clarity, and in the next section, a comparison between the U.S. and Korea will be made to examine the permissibility and limitations of the Korean case.

### 4.1. Calvary Chapel Dayton Valley v. Sisolak, 140 S.Ct. 2603 (2020)
4.1.1. Facts

Following the outbreak of COVID-19, 2020, the state of Nevada declared that indoor worship must be kept to a total of 50 or fewer people, whereas secular facilities, such as casinos, indoor gatherings, were allowed to continue operating so long as they only admitted 50% of their facility's fire-code capacity. Accordingly, religious groups contested the standard of 50 people or less and applied for an injunction. However, the Federal District Court of Nevada (Valley v. Sisolak, 2020) and the Federal Court of Appeals (Valley

v. Sisolak, 2020) dismissed the religious group's application. Accordingly, the case was referred to the Federal Supreme Court.

### 4.1.2. The U.S. Supreme Court's Opinion Regarding the Claim of Unconstitutionality

The first clause of the amendment, Freedom of Religion, requires that the law be textually neutral and generally applicable (Church of the Lukumi Babalu Aye, Inc. v. Hialeah 1993, at 531).

However, Paragraph 11 of the district decree 38–2 (ECF Doc. 38–2, §11) of this case does not limit worship to 50 or fewer people while allowing other secular gatherings. That is, Article 20 (Id. §20) of the Governor's decree relates to bowling, Paragraph 26 (Id. §26) relates to fitness, and paragraph 28 (Id. §28) relates to casinos. If these facilities scale down to 50% of the maximum occupancy, they are allowed to operate.

Casinos in the state of Nevada attract people from all over the country. Casinos serve alcohol and it would be unusual for patrons to keep 6 feet apart from one another. Similarly, church services at that time typically required believers to wear masks and encouraged people to maintain or exceed 6 feet between seats. Furthermore, church groups limited the worship time to 45 min. Nevertheless, to treat religious worship differently from casinos is to give preference to casinos and show disdain towards worship. In terms of uniform law enforcement, there is a difference between 50 or fewer people and reducing admittance to 50% occupancy. However, a different treatment of religion cannot be justified, since no sufficient reason can be admitted to support the difference.

A district ordinance restricting worship must be judged through a strict review. Therefore, the state must prove an indispensable interest for treating casinos and worship differently. However, the state failed to prove this. The state of Nevada also offered economic and public health factors as a justification for the discriminatory treatment of religious worship, but it was not persuasive for the following reasons:

First, the state of Nevada did not provide sufficient public health justification for taking a looser approach to restaurants, bars, casinos and gyms and a more stringent approach to places of worship. In other words, while states have discretion to treat entities differently in emergencies, they failed to demonstrate public health facts that could justify the imposition of strict restrictions specific for places of worship.

Second, it is for the state to balance individual economic hardship with public health concerns. Almost every state and municipality in the U.S. struggled to strike that balance. The best course of action for public health could lead to huge economic losses. However, it is not permissible to discriminately separate religious activities from other secular enterprises on the ground that religion does not generate sufficient economic benefits. Because states have a duty and responsibility to combat COVID-19, courts must respect state judgments. However, the state has a constitutional red line that cannot be crossed even in an emergency. This includes racism and religious discrimination (Calvary Chapel Dayton Valley v. Sisolak 2020).[7]

Third, the Nevada state government treated bars, casinos, gyms and worship services as separate items. However, state governments failed to prove that gatherings in bars and casinos were safe from a public health standpoint.

### 4.2. Roman Catholic Diocese of Brooklyn, New York v. Andrew M. Cuomo, 592 U. S. (2020)

#### 4.2.1. Facts

On 6 October 2020, in response to the spread of COVID-19, the Governor of New York promulgated an executive order and set the number of people who can participate in houses of worship in a certain area according to the zone (New York State Executive Order 202.69).[8] The number of people were capped at 10 or 25, depending on group size. According to the order, all non-essential gatherings were prohibited in red zones, all non-essential businesses were required to reduce in-person workers by 100%, and restaurants were only open for takeout or delivery services. However, the governor's executive order did not place restrictions on other businesses or gatherings and schools could continue

to conduct in-person classes. However, chapels were limited to 25% of their maximum occupancy or to 10 people, whichever was less. In orange zones, 'non-essential gatherings' were limited to 10 people, and gyms and tattoo shops with a high risk of infection were ordered to close. Restaurants could operate outdoors, and schools were obligated to conduct inspections, but in-person classes were possible. Again, chapels were regulated separately and only permitted "33% of maximum capacity or 25 people", whichever was the smallest. Additionally, in yellow zones, 'non-essential gatherings' were capped at 25 people, restaurants could operate indoors and schools were left open. However, chapels were limited to 50% of maximum capacity.

The order did not apply to "essential businesses"; that is, businesses seen as providing products or services necessary to maintain the health, welfare, and safety of New York State citizens. This included hospitals and grocery stores, as well as bars, pet shops and financial institutions (Executive Order 202.68, Empire State Dev. 2020)[9] The Governor asserted no evidence to support the procedure and its distinctions that determined whether businesses were essential. Nor did it assert that the distinction was based on an assessment of the risk of COVID-19 infection (Israel, 2020, at 7–8). Meanwhile, the Roman Catholic Diocese of Brooklyn comprises 210 churches, and in 2019 this would mean that a combined 1000 Masses would have been held on Sundays, with an average of 230,000 people attending. On 16 March 2020, Mass was suspended prior to the week's lockdown and all churches were closed on the 20th. In the meantime, the church consulted with medical professionals to review a safe way to hold Mass. After June, activities resumed with anti-infection measures in place, and weekend Masses were held in July, but the limit of 25% capacity was imposed even after the state lifted the same type of restriction elsewhere. The large-scale outbreaks of those churches is still unknown (Israel, 2020, at 11–12).

4.2.2. The Supreme Court's Collective Opinion (Per Curiam)

On 25 November 2020, the Federal Supreme Court upheld the application for emergency relief by collective opinion (per curiam) (Roman Catholic Diocese, 141 S. Ct. at 63, 63). According to this, the applicant clearly had the right to seek relief during the pending appeal. In other words, the claim of the First Amendment seemed likely to prevail, denying relief causes irreparable harm, and granting relief does not harm the public interest. The details were as follows (Roman Catholic Diocese, 141 S. Ct. at 63, 67–68):

①    The Supreme Court stated there was a problem with the neutrality and general applicability of the district decree. In other words, it was possible that the governor's executive order targeted Judaism (Stack 2020).[10] Moreover, it could not be considered neutral in that it treated chapels particularly strictly. Because it was in a red zone, synagogues and temples of Judaism were limited to 10 people, but 'essential businesses' were not limited. However, essential businesses included all production plants and transport, as well as campsites and garages. Additionally, in orange zones, chapels were limited to 25 people, but non-essential businesses were not. As a result, hundreds of people were permitted to shop at large retail outlets, while worship services were limited to 10 or 25 people. The governor stated that factories and schools contributed to the spread of COVID-19, but they were treated less harshly than the Diocese churches and Agudath Israel's synagogues, both of which demonstrated admirable safety records (Roman Catholic Diocese, 141 S. Ct. at 63, 67). The governor's executive order recognized essential interests, but the means were not appropriate (Roman Catholic Diocese, 141 S. Ct. at 63, 69).

②    It was recognized that religious freedom could not have been exercised. It was possible to watch worship services online, but doing so is not the same experience as attending in person. Catholics who attend Mass remotely cannot receive communion, and there are important religious traditions in the Orthodox Jewish faith that likewise require personal attendance. (Roman Catholic Diocese, 141 S. Ct. at 63, 68).

③    Finally, it had not been shown that granting the pending application would have harmed the public. The state had not claimed that attendance at the applicants'

services would have resulted in the exacerbated spread of the disease. Additionally, the state had not shown that public health would be imperiled if less restrictive measures had been imposed. (Roman Catholic Diocese, 141 S. Ct. at 63, 68).

### 4.3. Evaluation: Perspectives and Characteristics of Legal Interpretation in U.S. Cases

In the early stages of the state of emergency, the Supreme Court did not take into account the state's insufficiency of time, information, or expertise. However, as medical data became available, flexible amendments to state regulations were required. The Supreme Court required the state government to lift the stay-at-home order based on accumulated scientific knowledge. In the case of presenting the criteria for business resumption, the Supreme Court called upon the state government to clearly present the contents of the back-up plan to prevent reinfection, as well as regulations and the conditions necessary for release. The government was responsible for explaining where valuable human, time, and financial resources should be used among multiple policies. In court, the clarity of the standards for promulgating state decrees was an issue—whether it unreasonably restricted the rights of the First Amendment, whether the state government discriminated against a specific group without a reasonable reason, and other such matters. Based on these standards, different judgments had been made along the time axis with the issue of restrictions on religious freedom.

Specifically, the state was obliged to clarify the stay-at-home order and the criteria for lifting it. When a state presented its COVID-19 policy while using natural disaster and emergency provisions, they were required to demonstrate accountability and openness in decision-making to account for specific restrictions on citizens. New York State's quarantine measures can be highly appreciated in that they declared the separation of politics and science and accordingly implemented measures based on scientific data. According to the change in the infection rate and the number of patients, the degree of regulatory rigor changed, such as by imposing or lifting restrictions or allowing the reopening of business by classifying regions by color. However, the Supreme Court found that the classification and regulation of designated areas (to suspend indoor religious activities entirely) was still insufficient. The state government periodically revised the basic policies and checked their effectiveness. Accordingly, the Supreme Court required the government to change certain policies and explain the necessity and feasibility of changing those policies if it neglected to check existing policies. Federal courts judged on a time axis whether policy changes had been delayed due to insufficiency of government policies or insufficient rationale. Nevertheless, at present, it is difficult to reconcile the ongoing situation with a precise analysis that requires a sufficient degree of timeliness.

Based on these characteristic legal and institutional interpretations in the U.S. cases, we can learn the following facts: (1) In relation to the enforcement of the law, each ordinance restricting worship was judged through strict review; (2) The state was called upon to prove an indispensable interest in imposing stricter restrictions on certain religious groups over others; (3) It is up to the state to balance individual economic hardship with public health concerns, in terms of the balance between basic rights, such as minimizing irreparable harm, and public interest, as well as between neutrality and general applicability; (4) The state had to honor constitutional red lines that could not be crossed even in an emergency, such as engaging in racism or religious discrimination, as either would trigger intervention from the supreme court.

As such, considering this process of strict policy deliberation and the avoidance of constitutional red lines, the state–civil society relationship in the U.S. seems to be closer to Model 2 or 3 in that it is more systematic than that of the confrontational scheme shown in Model 1.

## 5. The Characteristics of Korea's Case and Issues of Institutional Complementation under Comparative Review with the U.S. Case

With respect to the above-described characteristics of legal interpretation on freedom of religion in the U.S., the characteristics and need for institutional complementation in Korea can now be outlined.

### 5.1. The Characteristics of the Korean Case Compared to the U.S. Case

In the U.S., religious freedom receives special treatment compared to other basic rights, whereas in Korea, religious freedom tends to be treated equally with, or is even less valued than other basic rights. To put this another way, in the U.S., because religious freedom is a basic and fundamental right, its restrictions can only be achieved if strict requirements are met. From the early stages of the COVID-19 outbreak, numerous complaints were filed and lawsuits still continue. On the other hand, in Korea, due to the *Shincheonji* church incident, which was believed to have exacerbated the spread of COVID-19, the government's regulation of religious organizations was relatively well accepted by the general public. However, as the government's comprehensive regulation continues, Korean churches and civic groups, such as those in the U.S., began to resist the government's excessive measures. Apart from the *Shincheonji* incident, what other factors contributed to this difference?

First, this difference may be related to the difference in the formation process of civil society undergone by the two countries. Compared to American civil society, which regards liberal values, such as religious freedom, almost as golden rules, Korean civil society was formed in the process of resisting excessive control by Confucian culture, Japanese colonial rule, and military dictatorship. In this process, during national crises, such as war or resistance against Japanese colonial rule or military dictatorship, as well as economic depression (e.g., the IMF) and large-scale epidemics (e.g., SARS), the state's power of governance was severely weakened. During such times, religious and social organizations played an active role in maintaining the national community by organizing movements of voluntary solidarity and resistance.

In particular, in Korean civil society, even though various religions, such as Buddhism, Confucianism, Christianity, Cheondogyo, and folk religions, functioned as a means of national governance in peacetime, when the community was in crisis, they played a conspicuous role as a spiritual pillar in the lives of citizens. As such, religions greatly contributed to the protection of social communities and the lives of individuals not only through civic movements, but also through the dissemination, transmission and education of collective social experiences or values. In other words, unlike in the case of the U.S., religious freedom in Korea in the early stages of the epidemic has served as a means for social community, and as a result, restrictions on religious freedom were taken for granted in the face of the unprecedented communal crisis that was COVID-19. Under such a crisis, an attitude that prioritized the interests of the community was requested through efforts such as righteous army activities (against Japanese imperialism right before the annexation of Korea), the independence movement of March 1 (during the Japanese colonial rule), gold-collecting campaigns (during the IMF period), as well as voluntary mask wearing and the practice of increased hygiene (during SARS). Here, we might be able to find certain elements of Confucian or communitarian socio-cultural experiences that have accumulated within the context of East Asian history wherein different forms of civil society led by religious groups played important roles.

On the other hand, when government restrictions are excessive and resistance to dictatorship becomes necessary, freedom advocacy by religious and social groups likewise played a leading role in shaping public opinion regarding civil liberties and other interests. As such, these differences in the formation process of civil society between the U.S. and Korea may be related to the characteristics of liberal and communitarian tradition, respectively. From this perspective, it can be seen that in the communitarian tradition the community itself, often centered on religion, social groups, and civic groups, plays signifi-

cant roles in protecting the rights and interests of the people themselves. This may partially explain the initial difference, as well as the two civil societies' subsequent responses to the excessive regulations implemented by their governments under similar COVID-19 pandemic situations.

Second, in Korea, the Constitution guarantees individual religious freedom based on the principle of separation of politics and religion, which is unlike the U.S., where religious freedom can also be restricted in accordance with Article 37 (2) of the same Constitution. When some Korean Christian groups conducted religious gatherings, especially Sunday worship services, at the risk of spreading contagions possibly resulting in mass infection, their actions were based on a claim that religious freedom, a fundamental right of the people guaranteed by the Constitution, had been violated. However, the government imposed suspensions on these religious activities, judging that COVID-19 threatened the people's right to health and national security. Yet, only after the initial phase, did challenges begin to arise around the fact that it was inappropriate to suspend Church gatherings while still permitting people to gather in large restaurants, public baths, and large department stores.

On the other hand, in the U.S., from the early stages of COVID-19, the main issue of conflict between state regulation and religious freedom under the First Amendment had to go through strict review according to detailed standards. When applying relaxed review standards under exceptional circumstances, they were divided into "determination of factual relations" and "constitutional evaluation of their restrictions", and their restrictions were required to meet "generality" and "neutrality". Only then could the relaxed examination standards potentially be applied. These review standards in the U.S. are in line with those of Korea based on the principle of excessive prohibition (proportionality). With this principle, the legitimacy of the purpose and the suitability of the means can be examined. At the stage of minimal intrusion, the standards may be relaxed to examine whether it is necessary to enforce certain restrictive measures. However, in the case of Korea, selective application according to context, such as the consideration of the minimum level of infringement, appears to be comparatively rare.

Third, in Korea, based on the government's quarantine guidelines, local governments took individual and specific measures to suspend in-person worship, and as a result, freedom of religion was more severely violated than other public facilities, and this was achieved without scientific or objective data. As such, the quarantine measures prohibiting in-person worship, or limiting the number of people to up to 50, without limiting the total number of people entering business places such as restaurants, can be seen as implying an error from ascertaining the facts. In contrast, in the U.S., the government's suspension of in-person worship required a 'constitutional evaluation' of the 'contents of regulation' even in situations where the confirmation of the 'factual relationship' and the resulting regulation were necessary. The determination of the facts here was to determine whether the spread of infection through in-person worship existed. For example, certain forms of in-person worship that require loud vocalizations, could be considered to present a heightened risk of infection. Yet, when the infection rate or the number of infected people does not reach the highest risk level, and when worship participants observe the required quarantine measures, it can be considered as low risk. From this point of view, general in-person worship itself cannot be directly equated with a high risk of infection.

Overall, the Korean government's restrictions on religious freedom and the enforcement of remote worship was not sufficiently reviewed. Furthermore, there were countless ways to achieve similar ends without having to excessively restrict religious freedom. From a legal point of view, there are several aspects that can be viewed as a form of coercion, which violates the principle of excessive prohibition and infringed upon the religious freedom guaranteed by the Constitution. Now, possible complements with reference to the U.S. cases can be considered.

### 5.2. Complementary Points for Korea's Quarantine Measures

As outlined above, in the early stages of COVID-19, there was a difference in Korea and the U.S. in terms of the public's reaction to excessive government regulations. In the case of the latter, the government's excessive regulatory measures could not be accepted uncritically. However, for the former, due to the *Shincheonji* and other church incidents that were suspected of having caused the spread of the Coronavirus, the government's regulation of religious organizations was relatively well accepted. However, as the government's comprehensive regulation continues, Korean churches and civic groups, like those in the U.S., began resisting the government's excessive measures.

As a result, the government's quarantine measures have changed significantly since the outbreak of COVID-19. In the beginning, the information, time and expertise to respond to the rapidly spreading crisis were lacking, but following gradual increases in available medical data, the requests for review and flexible application of regulations have also increased. In this respect, a rationale for further complementation would be that limitations on religious freedom should be kept to a reasonable and necessary minimum based on scientific knowledge according to the premise that religious organizations themselves make their best efforts to cooperate with the government's preventive measures.

In this regard, now the government should clearly define the stay-at-home order and the standards for lifting it while providing the rationale for the quarantine measures in detail and disclosing the decision-making process.

As for the problem of ascertaining the facts, resulting in the suspension of in-person worship without limiting the total number of people entering business places, the U.S. Supreme Court provided its opinion regarding the unconstitutionality claimed in Calvary Chapel Dayton Valley v. Sisolak (2020), and Roman Catholic Diocese of Brooklyn (2020). These cases can be taken into account. For instance, pursuant to these cases, restrictions on the number of participants in proportion to the size of the church can be more flexible. Especially, like in the latter case, "the maximum attendance at a religious service could be tied to the size of the church".[11] In that way, the challenged restrictions could be evaluated as "neutral" and of "general applicability", which would, thereby, satisfy the condition of "strict scrutiny", and not just end up being "narrowly tailored" measures that serve a "compelling" government's interest. (Roman Catholic Diocese, 141 S. Ct. at 63, 70)

As stipulated in the case of the Roman Catholic Diocese of Brooklyn (2020), the question of irreparable harm to the great majority of those who wished to attend Mass on Sunday (Roman Catholic Diocese, 141 S. Ct. at 63, 67–68) and others in similar situations could also be more carefully taken into account. Since the religious need to be a part of the holy space through direct communion with God, especially by participating in the essential rituals or gatherings, constitutes the very essence and spirit of religious freedom as a fundamental right. At the internal level, confession of faith is an area that cannot be violated, but religious acts, that is, acts of worship, belong to the realm of relative freedom of faith and can be restricted for the maintenance of order and public welfare. However, if confessions of faith are realized through a necessary form of community worship, such as the Sacrament, oblation, rites of baptism, etc., then external worship and internal confessions of faith can be viewed as an inseparable act. As noted in Roman Catholic precedents, prohibiting this without a legitimate reason would irreparably harm people's rights to happiness through religious activity. Therefore, according to this interpretation of the constitutional spirit and ethics, just as the inner confession of faith is absolutely protected, so the external worship needs to be protected accordingly. However, the Korean government treats churches the same as some indoor sports facilities, entertainment facilities (collatechs, clubs, entertainment pubs, etc.), internet cafes, karaoke rooms, and private institutes, and prohibits their operation. Since this is highly likely to infringe on religious freedom, it is required to treat churches differently from general indoor facilities.

Further, unlike the court decision on the Seoul Metropolitan Government's actions, if a COVID-19-related policy or legislative measure is made unilaterally or flows in a direction that excludes non-compliant minorities who are placed in a relatively vulnerable

environment or condition, the unconstitutionality of relevant laws and administrative measures can be judged more strictly in light of a specific situation or context.

Overall, as in the case of the U.S., which respects religious pluralism as a core right of freedom, in examining the scope and illegality of religious practices in Korea, a more stringent screening and flexible approach to application are required. There should be an aim for a robust pluralistic religious governance based on cooperation and harmony with the religious sphere of society.

## 6. Conclusions

The Korean government's quarantine measures were evaluated as a successful model to such an extent that it was branded 'K-Quarantine', but on the other hand, unlike the case of the U.S., certain aspects of that model can also be viewed as a serious threat to the religious freedom given that they neither obtained the consent of the public for governmental regulation nor respected anyone's right to raise objections and protests.

In the early days of the pandemic, the applications to repeal or reduce suspension filed by Korean Protestant churches claiming the suspensions of in-person worship were unjust and were rejected. The content of the ruling was that the public benefit gained by the prohibition was greater than the damage caused by the suspensions. However, in 2021, about a year later, the church's application to suspend the government's quarantine measures, which completely prohibited in-person worship, was cited. Following this decision, there was an increasing awareness that the government's measures, which even moved to close religious facilities without strict review, were in violation of the constitutional principle of excessive prohibition, which states that any violation of the legitimacy of the end, the suitability of the means, the minimum of infringement, and the balance of legal interests are unconstitutional. Nevertheless, compared to the U.S. and Western countries, more detailed legal interpretation and institutional supplements are needed to minimize the intervention of public authorities in the realm of civil society, especially that of religious institutions.

In this regard, in addition to the above-mentioned need for rigor in deliberation over legal regulations, it is also required to routinize the use of consultation bodies between civic groups and the government. This is because, unlike the U.S., there were many cases in which consultation was conducted in the form of notification right before the complete shutdown of religious activities in accordance with the law, rather than conducting multilateral dialogues on the level of equality and reciprocity. Seen from these points, compared to the U.S., which seems to be closer to Model 2 or 3 in its state and civil society relationship, Korea appears to be closer to Model 1 or 2 and should aim to move closer to Model 3.

In liberal democratic tradition, since Locke's division of 'religion' and 'politics', 'religion' was not understood as being subject to any human power; that is, here freedom was deemed necessary to value and respect in all circumstances. Situated in the realm of civil society, churches must cooperate and follow the government's policy, but in order to curb the government's excessive restrictions on religious freedom, it is also necessary to guarantee a more sophisticated "legal and institutional norm to make civil society more integrative and flexible through institutionalizing a new paradigm" (Yoo and Suh 2022, p. 2). If not, as Fitzgerald acknowledges, Locke's emancipatory account of the division of religion and politics would be likely to "work for a liberalism that serves ultimately to justify an oppressive capitalist state order", or as Israel argues, "religion can be managed by governments in a way that distracts from the pursuit of justice" (Stack et al. 2015, p. 18).

In this respect, even if some of the basic rights of the people may be restricted for the maintenance of order and the public welfare, more careful consideration is required, as in the case of the interpretation of the Constitution in the U.S. that religious freedom or human rights restrictions should be kept to a minimum. In this way, the government can be more respectful to religious diversity within a broader secularity of civil society as a principle of religious governance (Yoo and Suh 2022, p. 5). As Bader argues, "all regimes of religious governance lead to a certain institutionalisation and also require some forms

of public (administrative, political, legal or constitutional) recognition of religions. Public registration (e.g., for granting tax and other exemptions) requires criteria and thresholds in terms of numbers, time and durability of settlement, minimal stability and credibility, etc. Because it is inevitably selective, the rule of law minimally demands some forms of judicial control of administrative discretion". (Bader 2007, p. 294)

It is true that COVID-19 has caused an unusual crisis. Even so, considering the above points, even under these circumstances, it is essential to guarantee religious freedom as a fundamental constitutional right, and to protect the right to health for all citizens, including believers. Likewise, two years of trial and error on the part of the Korean government and religious groups give us a good reason to supplement more robust, viable, legal and institutional procedures to secure religious autonomy and diversity based on mutual respect between the state and religious groups. In addition, as Yoo and Suh insinuate, with this institutional system, which is based on equal and pluralistic value, it could also effectively "restrict an arbitrary demonization of religious organizations by government officials and politicians who value their political interest over religious diversity". (Yoo and Suh 2022, p. 6) Equally, believers, especially religious leaders, should also be keen on preserving freedom of conscience and cooperation with the state and society to establish a harmonized religious governance based on the spirit of religious pluralism.

**Author Contributions:** Conceptualization, methodology, software, validation, formal analysis, resources, D.C. and T.K.; investigation, data curation, writing—original draft preparation, D.C.; writing—review and editing, supervision, project administration, T.K.; funding acquisition, None. All authors have read and agreed to the published version of the manuscript.

**Funding:** This research received no external funding.

**Institutional Review Board Statement:** Not applicable.

**Informed Consent Statement:** Not applicable.

**Data Availability Statement:** National Assembly Library of Korea https://www.nanet.go.kr/main.do; Korea Citation Index; Korea Citation Index https://www.kci.go.kr/kciportal/po/search/poArtiSear.kci.

**Conflicts of Interest:** The author declares no conflict of interest.

## Notes

[1] This paper utilized social contract theory (e.g., that of Locke) and the legal perspective. According to this criterion, the state of freedom in the state of nature prior to the establishment of a nation was classified as "freedom", and the state of freedom as a basic right after the formation of a nation through a contract and transfer of rights was classified as "liberty".

[2] The Declaration of Independence uses the term "liberty" in terms of innate human rights. It says: "all men are created equal, that they are endowed by their Creator with certain unalienable Rights, that among these are Life, Liberty and the pursuit of Happiness". Yet, the U.S. Constitution (including the First Amendment) uses the term "Freedom of religion", in the sense that the state has to respect the inalienable state of freedom that humans innately enjoy.

[3] Freedom is divided into political freedom and mental freedom; political freedom is referred to as 'liberty', whereas mental freedom is often shortened to 'freedom'. Political freedom is the exercise of one's rights without being subject to domination or coercion from the powers of kings, governments or societies. For example, it refers to civil liberties related to thought, belief, movement, choice of occupation, etc. On the other hand, mental freedom refers to the ability to choose one's own will without being constrained by the outside world. In other words, freedom, as an abstract philosophical concept, is a term that expresses 'active state', 'freedom to create', 'freedom in which human beings exist', and is an active state of self-realization. On the other hand, liberty expresses a 'negative state', including the meaning of 'freedom from' and 'freedom from the control of something'. Yet, in modern law, it is mainly used in the context of securing individual basic rights from state control in accordance with the law.

[4] We can consider some examples, beginning with Martin Luther's 'Reformation' after 1517, the Augsburg Reconciliation between Protestants and Catholics in 1555, the Thirty Years' War in Germany from 1618–48, and in 1794, the enactment of the <General Land Act> of Prussia, where a policy of tolerance toward sects has been taken. Vgl. Axel Freiherr von Campenhausen, Religionsfreiheit, in: (Isensee and Kirchhof 2009). *Handbuch des Staatsrechts der Bundesrepublik Deutschland* VII, S. 600ff.

[5] Korea's Constitutional Court Decision 2000 Hun-ma 159 (27 September 2001).

[6] There are two clauses in the First Amendment guaranteeing freedom of religion: (1) The Establishment Clause prohibits the government from passing legislation to establish an official religion or preferring one religion over another; (2) The Free Exercise

Clause prohibits the government, in most instances, from interfering with a person's practice of their religion. Resistance to COVID 19-related government regulations is about violating the second clause.

7    New York State categorized regions by color into red, orange, and yellow zones based on the risk of COVID-19 infection. [New York State Executive Order 202.69, Continuing Temporary Suspension and Modification of Laws Relating to the Disaster Emergency (7 March 2020)].

8    The Governor's orders are posted at https://www.governor.ny.gov/executive-orders (accessed on 31 August 2022).

9    See Essential Business Guidance Related to Determining Whether a Business Enterprise Is Subject to a Workforce Reduction Under Executive Order 202.68, Empire State Dev., https://esd.ny.gov/ny-cluster-action-initiative-guidance (updated 15 December 2020).

10   Judaism could face fines for hosting thousands of weddings in New York (Liam Stack, Organizers of Wedding Fined for COVID Laxity, The New York Times, 25 November 2020).

11   The unconstitutional opinion of supreme court is as follows: "Almost all of the 26 Diocese churches immediately affected by the Executive Order can seat at least 500 people, about 14 can accommodate at least 700, and 2 can seat over 1000. Similarly, Agudath Israel of Kew Garden Hills can seat up to 400. It is hard to believe that admitting more than 10 people to a 1000-seat church or 400-seat synagogue would create a more serious health risk than the many other activities that the State allows". (Roman Catholic Diocese, 141 S. Ct. at 63, 67); Please refer to Agudath Israel v. Cuomo, No. 20A90, 2020 U.S. LEXIS 5707, 2020 WL 6954120 (U.S. 25 November 2020).

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
