# Peer review of "Governmental Response to ‘COVID-19’ and Religious Freedom in Korea as Compared to the United States"

_religions, doi:10.3390/rel14020173_

Round 1
Reviewer 1 Report
This Paper well points out the supplementary points to the korean goverment's religious policy through discussion and comparision related to freedom of religion and goverment regulation in Korea and the United State based on historical facts during the covid-19 outbreak. I don't think there's any problems with publishing in the journal Religions.
Author Response
Thank you for your generous evaluation. In order to more clearly reveal the differences between the U.S. and Korean cases, the concepts of freedom and liberty were more clearly defined based on the natural law tradition of Locke and others. Furthermore, by setting three types of relations between the state and civil society according to this standard, it was emphasized that Korea's case needs more institutional complementation than the United States. Meanwhile, we also added an aspect where the difference in initial response to COVID-19 could be related to the difference between the liberal and communitarian traditions of the United States and South Korea.

Reviewer 2 Report
This is a very timely study on how governmental measures against the pandemic are potentially threatening human liberty, freedom, and autonomy, esp. in the realm of religion.
However, while the study reviews legal and policy measures regarding religious action in Korea, it does not have any analytical criteria on which one can define what religious freedom, liberty, and human autonomy means. It only posits the antagonist, hostile, and binary distinction between state and society/religion as the foundation of societal/religious autonomy, freedom, and liberty. This is a poor conceptual assumption given the fact that there are very different understandings of what constitutes the legitimate relationship between state and religion and what societal/religious liberty truly means. Liberty and freedom are certainly not referring only to the absence of governmental interventions. Therefore, it is the very crucial task for the study to set up an analytical framework about what religious freedom means and how it can be gauged.
Without such work, it regrettably states "For a better solution to complement the institutional deficiencies and governmental attitudes toward religions, we now review the case of the U.S. state government and the response of the Supreme Court to the issue of religious freedom during COVID-19." In saying so, it implies that what is not like the US constitutes what is deficient in Korea. This is hardly justifiable.
If the study were to have developed several criteria on which religious freedom can be measured/reflected (instead of the current, arbitrary comparison with the US case; and only some cases and not all cases in the US, for the matter), its comparison with the US would have not been necessary. Instead, readers should read more details about the Korean case itself.
Author Response
Thank you so much for your valuable comments and detailed explanations to complement our research. You asked us to add any analytical criteria to which one can define what religious freedom, liberty, and human autonomy means, and what constitutes the legitimate relationship between state and religion because liberty and freedom are not referring only to the absence of governmental interventions. In other words, we understood that you suggested us to set up an analytical framework about what religious freedom means and how it can be gauged.
* Response: “We utilized the social contract theory (e. p., that of Locke) and the legal perspective in line with the natural law tradition. According to this criterion, the state of freedom in the state of nature prior to the establishment of a nation was classified as “freedom,” and the state of freedom as a basic right after the formation of a nation through a contract and transfer of rights was classified as “liberty” (Footnote 1). Accordingly, “this paper tends to use ‘freedom’ in a more limited sense of the inalienable, ideal state of freedom that humans innately enjoy from a state of nature.’ Freedom of conscience, will or action can be pertained to this category. On the other hand, ‘liberty’ is used to refer to legal rights based on autonomy formed after the introduction of social contract or law to which individuals ceded power for their protection. Human autonomy refers to reason that can control decisions and actions by oneself, not by external coercion. From this perspective, this paper tends to use the term ‘religious liberty’, in the sense that it presupposes the citizen’s potential for autonomous religious actions within a legal national community.” (p.2)
Regarding this point, we also added a footnote 3, as well as 2, detailing the division between political freedom and mental freedom, while explaining that political freedom is referred to as ‘liberty’ and mental freedom as ‘freedom’. Based on these criteria, political freedom is the exercise of one's rights without being subject to domination or coercion from the powers of kings, governments and societies, such as civil liberties related to thought, belief, movement, choice of occupation, etc. On the other hand, mental freedom refers to the ability to choose one's own will without being constrained by the outside world. In other words, freedom, as an abstract philosophical concept, is a term that expresses 'active state', 'freedom to create', 'freedom in which human beings exist', and is an active state of self-realization. On the other hand, liberty expresses a 'negative state' including the meaning of 'freedom from' and 'freedom from the control of something'. Likewise, in this research, according to the tradition of modern law, liberty is mainly used in the context of securing individual basic rights from state control in accordance with the law.
As a result, we confined our discussion to the negative sense of liberty, referring only to the absence of governmental interventions. Yet, based on this conceptual definition, in order to reconcile the problem between the state and religion, we also suggested several models for the relationship between the state and religion within civil societies: 1) a confrontational structure between the two, (2) a symbiotic cooperative structure consisting of the state's protection of religion within civil society, and the voluntary cooperation of religion with the state, (3) Various models, including religious governance, whereby voices of civil society are reflected in the state through consultative bodies made up of government agencies, local governments, civic groups, interest groups, as well as religious groups and lay service organizations, such that the state can converge the opinions of religious organizations in connection with civil society. Under that model, religious groups can effectively influence state actions or policy decisions by utilizing these consultative bodies made up of various forms and channels. In general, mature democratic societies aim for models 2 and 3. (p.17).
In relation to this purpose, comparing the U.S. case does not mean that “what is not like the US constitutes what is deficient in Korea.” We also pointed out differences in the process of forming civil societies between the United States and South Korea and the resulting differences in their initial response to COVID-19. However, through a comparative analysis of the representative cases of the U.S. and Korea, the differences between the two countries were highlighted in the sense that, in the case of the United States, the government and the judiciary have stricter procedures and processes to deliberate on many factors according to scientific data before enforcing laws, while using various consultative bodies within civil society.
We just tried to show that in the U.S., the strict policy deliberation process, and the constitutional red line seems to be closer to the model 2 or 3 in its state-civil society relationship, in that it is more systematic than that of the confrontational scheme of model 1 (p.13). In the process of preparing the thesis, several more U.S. cases were prepared, but the process and content of the cases are almost the same, so we believe there is no problem in viewing the selected U.S. cases as representative precedents. [Due to the length of the thesis, other cases will be introduced in another thesis next time.]
Also, we also added the following limitations of this research at the end of the introduction: “Lastly, one limitation of this study that should be stated is that the sociological discussion of religion has not been sufficiently reviewed, and therefore, this thesis focuses on legal reflections on governmental restrictions on religious activities via the comparison of South Korea and the U.S."(p.2)
In addition, English proofreading was also completed for the entire thesis.
Thanks again for your constructive comments.

Reviewer 3 Report
Review
This paper entitled “Government’s Response to ‘COVID-19’ and Religious Freedom in Korea: A Comparative Review with the United States” offers a legal reflection on the quarantine policies by the comparison of the two countries.
It is a very well-written paper dealing with religion, state, and law, including excellent legal reviews on the quarantine policies and the freedom of religion. In particular, the authors conducted content analysis on legal documents very well; however, they need to take into serious consideration perspectives of the sociology of religion other than the legal viewpoint. In other words, the quarantine policies are all social constructions and they cannot be evaluated only by the court judgment. Therefore, it should be clearly stated as the limitation of this study.
1) “The freedom of religion” has different historical and socio-cultural implications in Korea and in the United States. It is the core value of American civil society, you may confer Alexis de Tocqueville; however, it has been developed in the conflict with the Japanese Colony and the Military dictatorship in the modernization process in Korea.
2) The outbreak of Covid-19 has been differently understood by the two countries. The effect of Shin-cheon-ji had the most significant impact on the super-spread of the pandemic in Korea, so the strict regulation or ban on religious gatherings was fairly accepted in the civil and religious arena in February/March 2020.
Thus, I don’t agree that “Both countries share a common view that people cannot uncritically accept the government’s excessive quarantine measures as in the early stages of the pandemic” (p13, lines 603-4). Authors need to face the different trajectories of the pandemic, especially at the very beginning of its spread, in different socio-cultural contexts of the two countries.
3) I’d like to offer a critical reflection on the authors’ premise that “a rationale for the necessary complementation would be that religious freedom should be limited to a reasonable and necessary minimum based on scientific knowledge.” As a matter of fact, “scientific knowledge is itself a contested output of social construction.” WHO did not admit the effect of wearing a (face)mask until July/2020; however, Koreans already learned its effect of preventing infection through collective experiences (see Pajoo, Starbucks case, May, 2020). Authors need to consider the wider contexts of society and the limit of unilateral legalistic interpretation.
You may discuss a “state of exception” as the effect of a global pandemic in which no scientific evidence “normally” functioned. Because of it, as you mentioned, “In the begging, the information, time and expertise to respond to the rapidly spreading crisis were lacking” (p13, line 610).
Author Response
Dear Sir,
Thank you for your valuable and constructive comments with detailed explanations. You pointed out that we need to seriously consider perspectives of the sociology of religion because “the quarantine policies are all social constructions and they cannot be evaluated only by the court judgment. Therefore, it should be clearly stated as the limitation of this study.”
*Response: We mostly agree that the quarantine policies are social constructions because they depend on the social context and circumstances of each country, but since this thesis focuses on a legal reflection on the quarantine policies by the comparison of the two countries, the sociological discussion of religion was not dealt with much. Accordingly, the following limitation was specified at the end of the introduction: “Lastly, one limitation of this study that should be stated is that the sociological discussion of religion has not been sufficiently reviewed, and therefore, this thesis focuses on legal reflections on governmental restrictions on religious activities via the comparison of South Korea and the U.S."(p.2)
Further, you pointed out that: 1) “The freedom of religion” has different historical and socio-cultural implications in Korea and in the United States. It is the core value of American civil society; however, it has been developed in the conflict with the Japanese Colony and the Military dictatorship in the modernization process in Korea.”
* Response: We mostly agree on this issue. Accordingly, in relation to the differences in the process of forming civil society, we paid more attention to the socio-cultural context of Korea's communitarian tradition in contrast to the American liberal tradition. From this perspective, we specified that the communitarian tradition in which the community itself, centered on religion, social groups, and civic groups, plays a significant role in protecting the rights and interests of the people themselves, may partially explain the initial difference, as well as the civil society’s subsequent response to the government's excessive regulation in the same COVID-19 pandemic situation. (p.14)
2) Your 2nd point is as follows: 2) (1) “The outbreak of Covid-19 has been differently understood by the two countries. The effect of Shin-cheon-ji had the most significant impact on the super-spread of the pandemic in Korea, so the strict regulation or ban on religious gatherings was fairly accepted in the civil and religious arena in February/March 2020. Thus, I don’t agree that ‘Both countries share a common view that people cannot uncritically accept the government’s excessive quarantine measures as in the early stages of the pandemic’ (p13, lines 603-4). (2) Authors need to face the different trajectories of the pandemic, especially at the very beginning of its spread, in different socio-cultural contexts of the two countries.”
* Response: (1) On the first sub-point, we also specified this fact at the beginning of this paper, but following your suggestion, it was emphasized as a key difference between the two countries during the initial response to the pandemic. (2) On the second sub-point, as described above, we explained the different trajectories in different socio-cultural contexts of the two countries, while specifying that unlike the U.S. case, “In Korea, due to the Shincheonji church incident, which was believed to have caused the spread of Corona, the government's regulation of religious organizations and the general public was relatively well accepted.” (p.14)
3) Your third point is as follows: (1) “I’d like to offer a critical reflection on the authors’ premise that “a rationale for the necessary complementation would be that religious freedom should be limited to a reasonable and necessary minimum based on scientific knowledge.” As a matter of fact, “scientific knowledge is itself a contested output of social construction.” WHO did not admit the effect of wearing a (face)mask until July/2020; however, Koreans already learned its effect of preventing infection through collective experiences (see Pajoo, Starbucks case, May, 2020). (2) Authors need to consider the wider contexts of society and the limit of unilateral legalistic interpretation. You may discuss a “state of exception” as the effect of a global pandemic in which no scientific evidence “normally” functioned. Because of it, as you mentioned, “In the begging, the information, time and expertise to respond to the rapidly spreading crisis were lacking” (p13, line 610).”
* Response: (1) We partly agree that scientific knowledge is itself a contested output of social construction, in that (i) the output can be changed according to the development of science, and (ii) scientific data can also change according to the researcher's desired purpose and values, which cannot be separated from the social environment and experience. In this sense, the discussion of collective experience can be quite persuasive, such that we partly stated this factor in our above discussion. (p.14)
Concerning the mask example, as you pointed out, even when the effectiveness of masks has not been scientifically proven, it seems that Koreans have learned empirically that masks have minimal deterrence against respiratory diseases caused by droplets through the experience of SARS. On the other hand, if the purpose of what you said is further expanded, even if the WHO acknowledged the effectiveness of masks at the later stage, it cannot be regarded as scientifically proven either. This is because there are many researches and discoveries that question the preventive power of masks and even discuss the dangers of the masks themselves.
(2) Regarding the last sub-point on the state of exception, our answers are as follows: In general, in the case of the people without professional scientific knowledge, there may be some cases that can admit state of exceptions regardless of scientific data, as Korean citizens protected themselves based on past experience, in which Korean civil society has taken the lead. However, when looking at the Korean case as a whole, as the reviewer pointed out, we pointed out that socio-cultural experiences accumulated in different historical trajectories worked, and in this process, civil society such as religious groups played an important role, rather than explaining this case as a state of exception.
Thanks again for your precious comments.
Sincerely Yours,
